# Analysis of Mouse Blood Serum in the Dynamics of U87 Glioblastoma by Terahertz Spectroscopy and Machine Learning

Denis Vrazhnov [1,2], Anastasia Knyazkova [1,2], Maria Konnikova [3,4], Oleg Shevelev [5], Ivan Razumov [5], Evgeny Zavjalov [5], Yury Kistenev [1,2], Alexander Shkurinov [3,4] and Olga Cherkasova [4,6,7,*]

1. Laboratory of Biophotonics, Tomsk State University, 634050 Tomsk, Russia
2. V.E. Zuev Institute of Atmospheric Optics of Siberian Branch of the Russian Academy of Sciences, Academician Zuev Square, 1, 634055 Tomsk, Russia
3. Faculty of Physics, Lomonosov Moscow State University, 119991 Moscow, Russia
4. Institute on Laser and Information Technologies, Branch of the Federal Scientific Research Centre "Crystallography and Photonics" of the Russian Academy of Sciences, 140700 Shatura, Russia
5. Federal Research Center "Institute of Cytology and Genetics of the Siberian Branch of the RAS", 630090 Novosibirsk, Russia
6. Institute of Laser Physics, Siberian Branch of the Russian Academy of Sciences, 630090 Novosibirsk, Russia
7. Faculty of Automation and Computer Engineering, Novosibirsk State Technical University, 630073 Novosibirsk, Russia
* Correspondence: cherkasova@laser.nsc.ru or o.p.cherkasova@gmail.com

**Abstract:** In this research, an experimental U87 glioblastoma small animal model was studied. The association between glioblastoma stages and the spectral patterns of mouse blood serum measured in the terahertz range was analyzed by terahertz time-domain spectroscopy (THz-TDS) and machine learning. The THz spectra preprocessing included (i) smoothing using the Savitsky–Golay filter, (ii) outlier removing using isolation forest (IF), and (iii) Z-score normalization. The sequential informative feature-selection approach was developed using a combination of principal component analysis (PCA) and a support vector machine (SVM) model. The predictive data model was created using SVM with a linear kernel. This model was tested using k-fold cross-validation. Achieved prediction accuracy, sensitivity, specificity were over 90%. Also, a relation was established between tumor size and the THz spectral profile of blood serum samples. Thereby, the possibility of detecting glioma stages using blood serum spectral patterns in the terahertz range was demonstrated.

**Keywords:** terahertz time-domain spectroscopy; machine learning; mouse blood serum; U87 glioblastoma

## 1. Introduction

Gliomas are the most common brain tumors. Glioblastoma multiforme (GBM) is the most aggressive, invasive, and undifferentiated type of glioma, and has been designated grade IV astrocytoma by the World Health Organization [1]. GBM is one of the most rapidly progressing oncological diseases and has unfavorable survival prognosis [2,3]. The main reason for this is late diagnosis [4]. Early glioblastoma detection is based on carcinogenesis molecular pathway discovery [5]. Traditional experimental methods, such as liquid chromatography and mass spectrometry [6–8], as well as nuclear magnetic resonance [9,10], cannot provide completely reliable information due to complex preparation of samples, long testing time, and the impossibility of timely intraoperative analysis. Early and noninvasive diagnosis of oncological diseases can be achieved using body fluid spectral pattern measurement [11–14]. The applicability of terahertz time-domain spectroscopy (THz-TDS) blood plasma analysis for diagnosis of liver cancer [15] and malignant thyroid nodule [16] has been demonstrated.

A glioblastoma animal model makes it possible to investigate tumor development under controlled conditions. Orthotopic xenotransplantation of human cells into immunodeficient animals is the most common model of glioma [17,18]. Currently, continuous cell lines such as U87 derived from primary human tumor cells (transplanted intracranially into the mouse brain) are widely used to create such models of human glioma [18,19].

The blood serum THz absorption spectra do not have sharp peaks [20,21]. In this case, differentiation of various groups can be achieved using machine learning (ML) [14,22–26]. The ML pipeline includes different methods to remove noise, select informative features, built predictive data models, etc. The sequence of algorithms depends on data peculiarities. Sharp noise peaks in THz spectra can be removed by proper smoothing [27–29]. The data homogeneity of every class is defined by the presence of outliers. Signal denoising and outlier removal are optional steps, applied when there are specific problems with data, such as data acquisition errors caused by hardware noise, or inhomogeneity of biological samples caused by individual characteristics of glioma development in different animals. Balance among classes influences the effectiveness of the predictive data model. Justification of the choice of appropriate algorithms can be confirmed by external validation procedures, i.e., ML predictive model scoring [30,31].

Small volumes and high dimensions of data are the major problems in medical studies. Dimensionality reduction is performed by feature extraction, which involves transformation of the original data to remove redundant data. Many feature-extraction techniques have different benefits; among the most reliable are loadings matrix analysis in PCA (LMPCA), and Shapley values [32,33]. PCA has proven its effectiveness in multiple studies of THz, IR, and Raman spectra [12,13,16,34]. Yet LMPCA is a linear method, which might not be effective for complex data but gives repeatable results. Shapley values can be used along with arbitrary predictive model-construction algorithms, but they are computationally costly owing to the high dimensionality of data.

Several ML methods have been used for small volume and high dimensional biomedical data, such as support vector machines (SVMs), gradient boosting (GB), and random forests (RFs) [16]. SVM is a robust algorithm allowing the construction of a non-probabilistic binary linear classifier, which maximizes the margin width between two classes in the feature space. The kernels can be considered as projective functions to map the input data to a higher dimensional space. To retain explicability, the data model should be linear. The linear kernel is the simplest, allowing construction of a direct informative feature-selection procedure based on the distance of the feature vector from the margin. The k-fold or random train–test split cross-validation (CV) model is commonly applied for estimating the performance of a predictive data models. The k-fold CV has reasonable computational load.

In this work, during experimental U87 glioblastoma development, mouse blood serum was studied by THz-TDS, and a ML-based predictive data model for glioblastoma detection was created.

## 2. Materials and Methods

### 2.1. Samples

The study was carried out in accordance with the EU Directive 2010/63/EU and the ARRIVE 2.0 guidelines, and approved by the Inter-Institutional Commission on Biological Ethics at the Institute of Cytology and Genetics, Siberian Branch of the Russian Academy of Sciences (Permission #78, 16 April 2021).

The model of orthotopic xenotransplantation of U87 human glioblastoma cells into immunodeficient SCID animals was used [18]. In brief, the U87 cell suspension was introduced in the subcortical brain structure through a hole in the animal's cranium. Animals from the control group were injected in a similar manner with 5 μL of the culture medium. The tumor size was measured using a horizontal 11.7 Tesla MRI tomograph (Biospec 117/16; Bruker, Billerica, MA, USA). the animals were immobilized by isoflurane (Baxter International Inc., West Deerfield Township, IL, USA) using the Univentor 400 Anesthesia Unit (Univentor Ltd., Zejtun, Malta). The animals were removed from the experiment by

decapitation on days 7, 14, and 21 after injection. Blood samples were collected in separate tubes and kept at room temperature for 45 min. Then tubes were centrifuged for 15 min at 1000 g. Serum was cast into separate tubes and frozen at −80 °C. Each experimental group had a corresponding control group. There were 10 mice in each group.

*2.2. THz Spectroscopy*

Analysis of the solutions (distilled water or blood serum samples) was carried out in special cuvettes, which were pre-printed on a 3D printer using Watson material (styrene butadiene copolymer, SBS) [35]. The selection of material for printing was made due to its sufficient transparency in the THz region of the spectrum [36], as well as high moisture resistance. When preparing for printing on the 3D printer, the standard preset provided by the "Polygon X" program was used, and the percentage of filling of model areas was specified as 100% (cast infill). Printing was carried out using fused deposition modeling (FDM). The principle of building according to FDM technology lies in the layer-by-layer growth of a product from a pre-melted plastic thread. Sample roughness is not automatically controlled. Cuvettes with obvious printing defects were not used for the study.

An example of an average terahertz spectrum from 10 empty cuvettes is shown in Figure 1. As can be seen, there was little variation. The absorbance of the cuvette was the same for all printed samples. THz transmission spectra of mouse blood serum samples were measured in the range 0.2 to 2 THz by a T-SPEC spectrometer (EKSPLA, Vilnius, Lithuania). The spectral resolution was 1 GHz, the acquisition rate up to 10 spectra/s, and the dynamic range better than 70 dB.

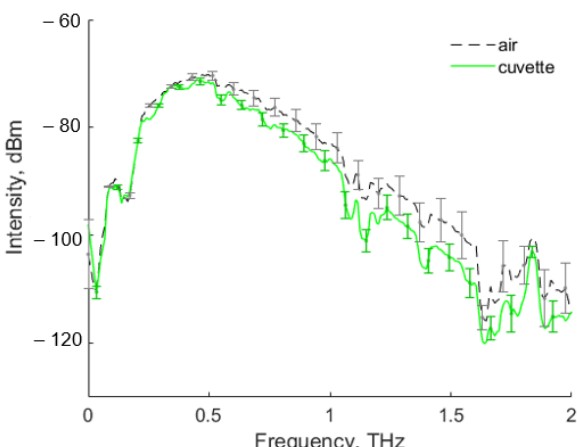

**Figure 1.** Example of an average THz spectrum from 10 empty cuvettes with standard deviation.

A cuvette was placed in the THz-TDS spectrometer so that the THz radiation beam passed through the cuvette center (see Figure 2a). THz radiation was focused onto the sample using parabolic mirrors. At the focus point, the THz beam diameter was about 3.5–4 mm. The thickness of the cuvette (Figure 2b) was 0.5 mm (Figure 2a). The position of the cuvette in the THz-TDS spectrometer is shown in Figure 2c. The measurements were carried out at a constant room temperature of 21 ± 1 °C.

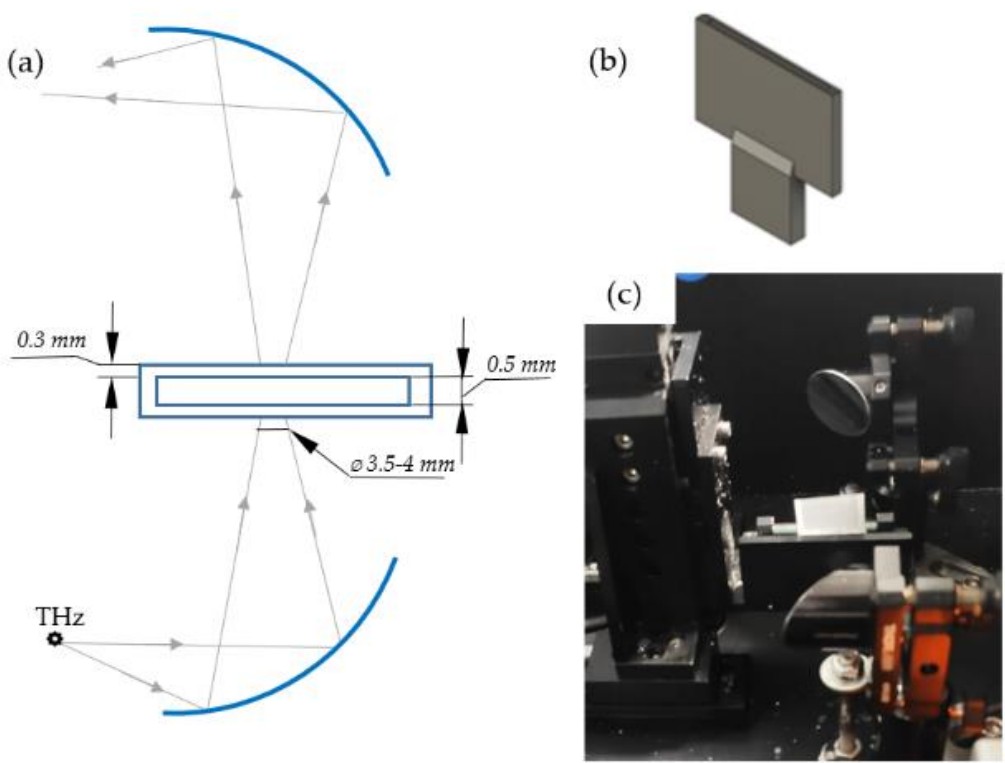

**Figure 2.** (**a**) Scheme of the experiments and parameters of the cuvette, gray arrows show the path of the THz radiation beam; (**b**) the cuvette printed on a 3D printer; (**c**) view of the cuvette installed in the THz-TDS spectrometer.

The THz signal registered from the empty cuvette was used as a reference. Then, without changing the position, the cuvette was filled with either distilled water or a pre-thawed blood serum sample with a volume of 50 μL, using an automatic dispenser. A new cuvette and dispenser tip were used for each sample. Sample spectral scanning was conducted along six spatial points of the cuvette with a step of 0.1 mm vertically and horizontally. Time averaging (over 256 spectra) was performed for each spatial point of the 2D scan. The signal transformation from a temporal to a frequency domain was conducted by TeraVil TRS-16 software (TeraVil Ltd, Vilnius, Lithuania). The impulse (time signal) and intensity spectra without a cuvette, with an empty cuvette, and for the cuvette with distilled water or blood serum sample are shown in Figure 3.

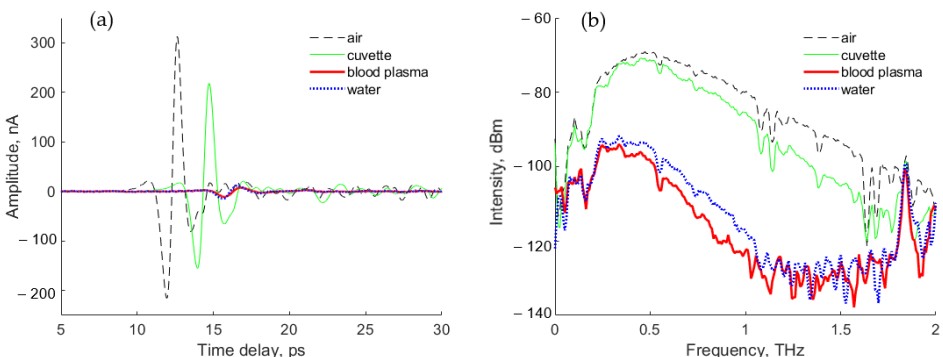

**Figure 3.** THz spectra in (**a**) temporal and (**b**) frequency domains: without a cuvette (black dashed line); with an empty cuvette (green line); with the cuvette filled with distilled water (blue dotted line); or a blood serum sample (red line).

### 2.3. Machine Learning Methods

Studied samples were split into two classes: tumor and control. To verify separability of the classes at each time point, we used the ML pipeline (MLP) presented in Figure 4.

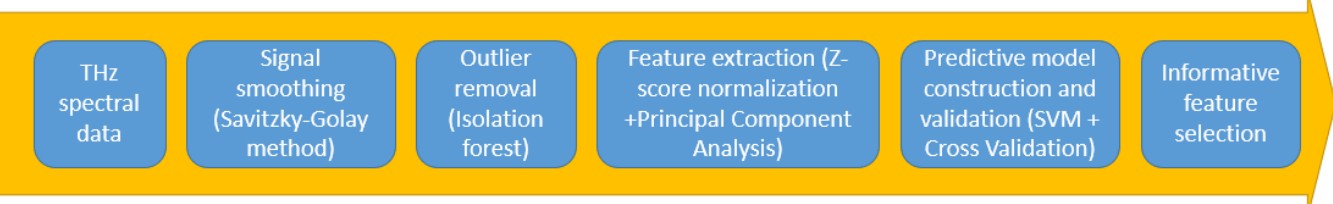

**Figure 4.** Proposed MLP for THz spectral data analysis.

In the first stage, THz spectra were smoothed by the Savitsky–Golay filter to remove sharp spikes, using the following parameters: filtering window width (windows_size) 15, degree of smoothing polynomial (polyorder) 2, and smoothing of the spectrum derivative (deriv) was not used. The parameter names were given according to the scipy.signal.savgol_filter library in the Python language. At the second stage, we implemented outlier removal by the isolation forest (IF) method [37]. IF is a machine learning technique, which uses an ensemble of binary decision trees called isolation trees (see Figure 5) to discover anomalies in data based on the distance of the current point in feature space from the others. Unlike other outlier removal methods, IF does not model normal points distribution to find deviations. A set of THz spectra belonging to the same class were used in the IF input. In the upper node of the isolation tree, a single component of the spectrum was analyzed to find the outlier. If it was not found, the process continued on the lower nodes. The decision to considered a feature vector as anomalous depended on the number of randomly chosen vector components required to isolate it from the other data. This algorithm was implemented in the Python language with the sklearn.ensemble.IsolationForest library. The isolation number parameter was chosen automatically [38].

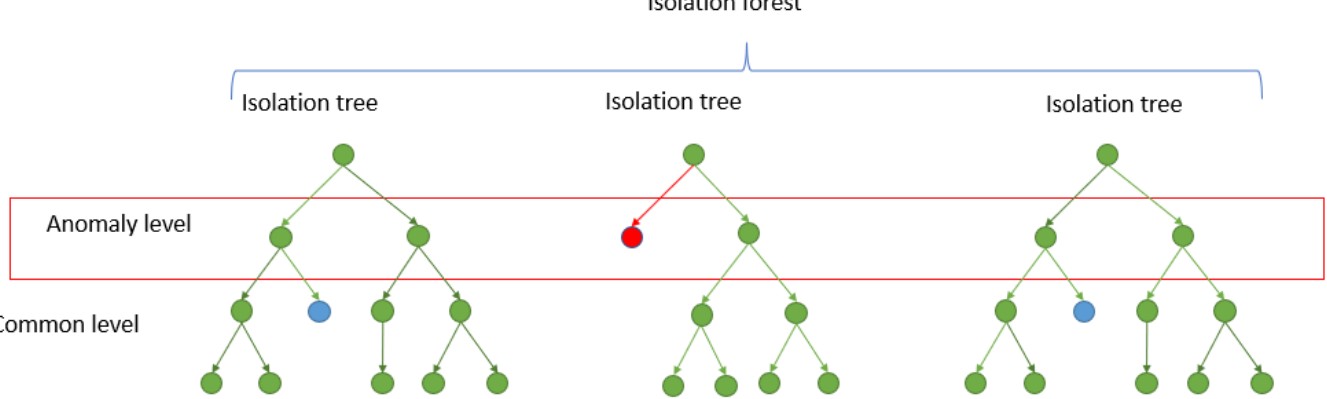

**Figure 5.** The concept of the isolation forest algorithm. Green dots are a common data; blue dots are rare data, but not an anomalous; the red dot is an anomaly. The closer an endpoint dot is to a tree root, the higher the anomality score. Each node is a randomly chosen component of the studied spectrum.

In the third stage, the data were standardized by subtracting the mean and normalizing to the standard deviation (Z-score normalization). PCA was chosen to construct a sequential algorithm for the informative feature selection. The number of principal components remaining were 10, to cover more than 95% of variance in data. The informative features were selected from the rows of the loadings matrix with the highest absolute values. PCA and Z-score were implemented using sklearn.decomposition.PCA and sklearn.preprocessing.StandardScaler Python modules. The separability of data was analyzed by the predictive model, based on SVM with linear kernel at the fourth stage.

The sklearn.svm.SVC Python module with default parameters was applied to construct the classifier. The k-fold CV was used. The algorithm was implemented in Python using the model_selection.StratifiedShuffleSplit module of the sklearn library. The quality of classifiers was estimated using the receiver operation curve (ROC) and area under curve (AUC) analysis. ROC plots a graph of sensitivity versus 1-specifity, and AUC displays the area under the ROC.

For the final stage, we developed a sequential feature selection algorithm by combining PCA and SVM (see Figure 6). Note that PCA and SVM are deterministic methods, so the difference in results can only be obtained by randomly splitting the initial dataset for the training and testing samples.

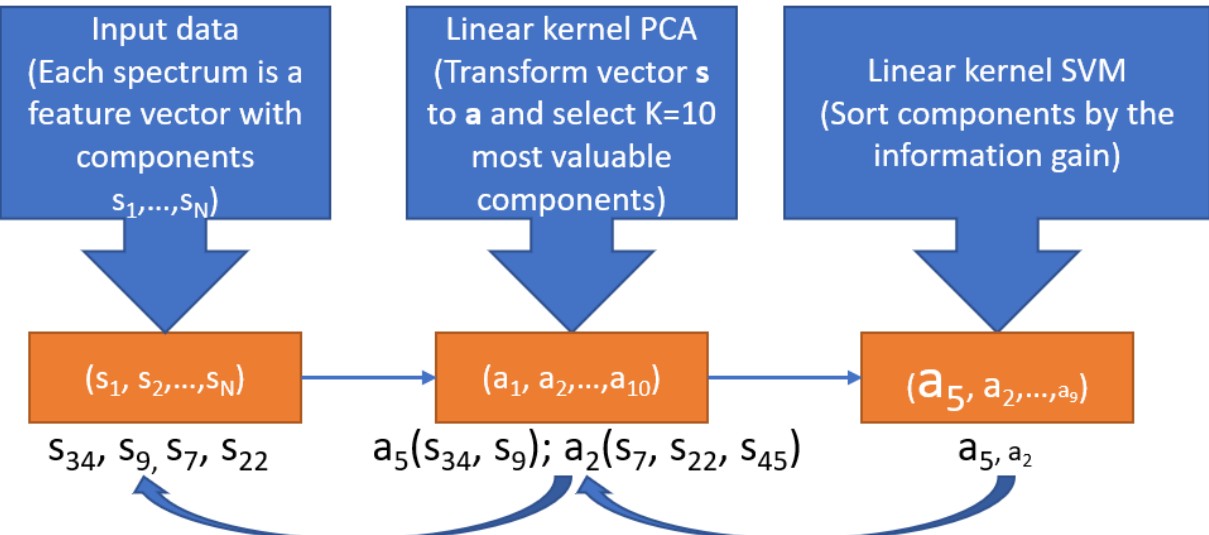

**Figure 6.** Sequential feature selection algorithm, based on PCA and SVM. Each input spectrum is considered as a multidimensional feature vector s, which is transformed by linear kernel PCA to the vector a with fewer components (K = 10 was used as an example). Transformed vectors are classified by SVM and each component is sorted according to influence on prediction accuracy (larger font size represents bigger influence). Components with the most influence are analyzed by a respective loadings matrix in PCA, and the most informative initial components are revealed.

The LASSO (least absolute shrinkage and selection operator) method of multivariate regression with $L_1$ regularization was chosen to investigate the relation between the THz spectra and the tumor size [39]. The linear LASSO model is described by the formula:

$$\min_w \frac{1}{2N} \|Xw - y\|_2^2 + \alpha \|w\|_1,$$

where $N$ is the number of samples, $X$ is the initial spectra, $y$ is tumor size, $w$ is regression coefficient, $\alpha$ is the regularization parameter, $\|w\|$ denotes vector norm, lower index 1 and 2 are the $L_1$ and $L_2$ norm, respectively. The non-informative features correspond to zero regression coefficients, the larger regression coefficients indicate high significance. These features correspond to the informative THz frequencies and allow narrowing of the THz interval for glioblastoma detection.

## 3. Results and Discussion

Increased tumor growth was observed from 14 to 21 days after U87 cell injection into the subcortical brain structure (see Table 1).

**Table 1.** U87 glioblastoma volume (mm$^3$). Values are presented as means $\pm$ standard error of the mean.

| Experimental Group | Days after Injection of U87 Cells | U87 Glioblastoma Volume, mm$^3$ |
| --- | --- | --- |
| U87-1 | 7 | $2.6 \pm 0.4$ |
| U87-2 | 14 | $10.6 \pm 1.8$ |
| U87-3 | 21 | $89.6 \pm 11.5$ [1] |

[1] $p < 0.001$.

Tumor growth causes a significant change of blood composition [9,11–13,15,16,40]. Serum THz absorption was shown to decrease when tumor size increased. Similarly, the U87-3 group had the smallest THz absorption among the groups [41].

### 3.1. THz Data Analysis

A pairwise comparison of THz spectra for experimental and corresponding control groups is shown in Figure 7. The non-overlapping region of 0.5–0.8 THz appeared promising for further analysis.

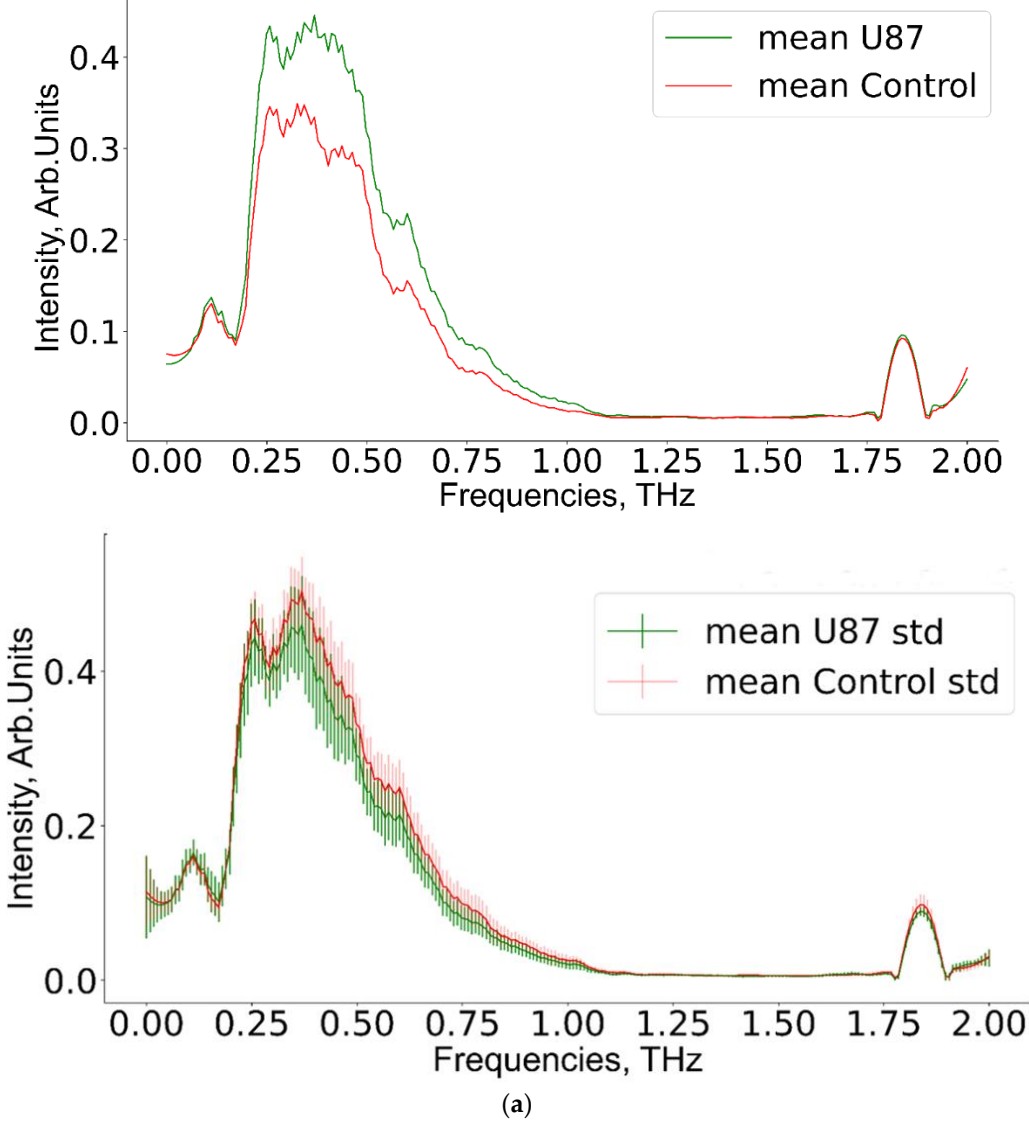

(a)

**Figure 7.** *Cont.*

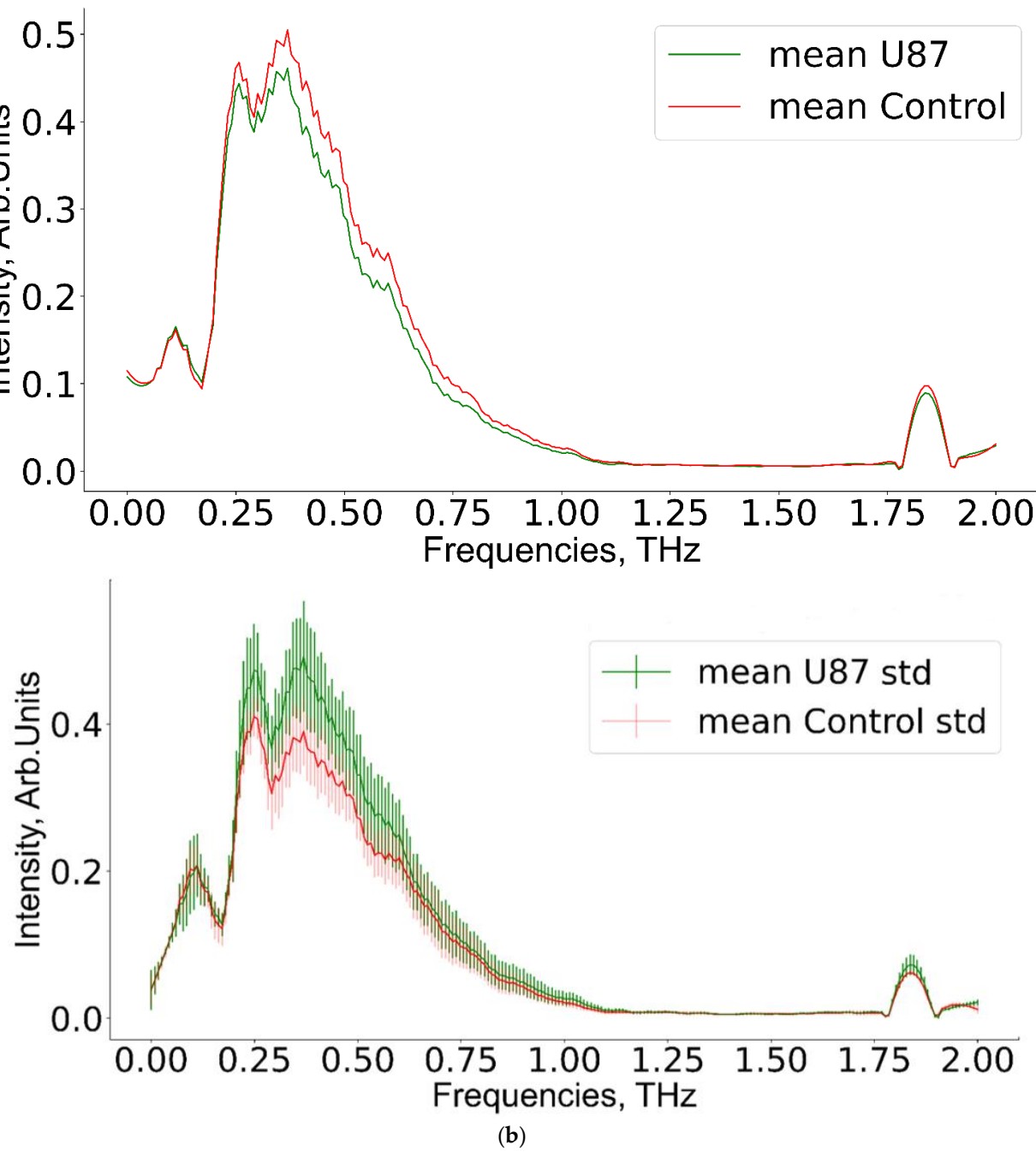

**Figure 7.** *Cont.*

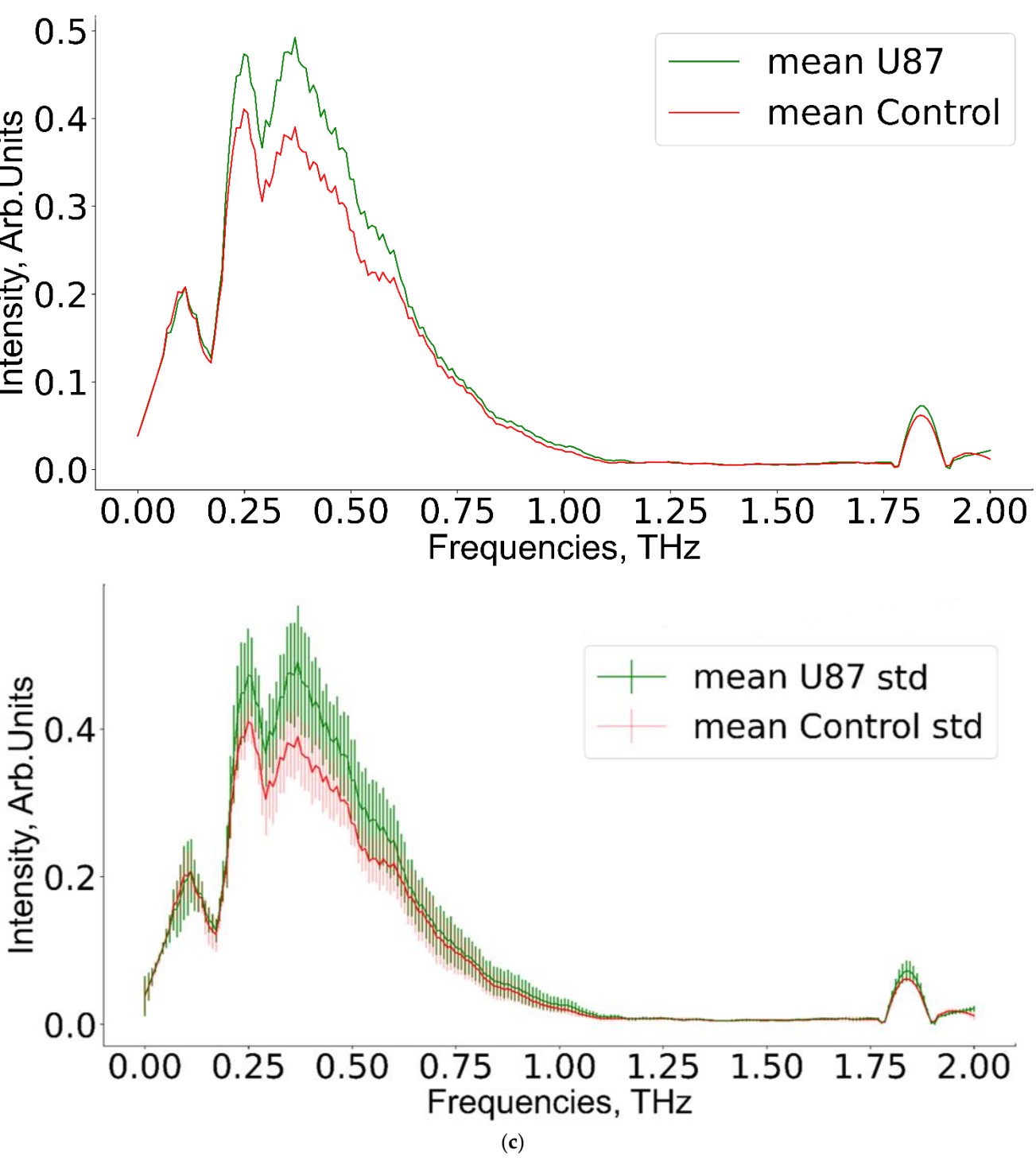

(**c**)

**Figure 7.** Mean THz spectra and scatter values for (**a**) experimental group 1, (**b**) group 2, (**c**) group 3, and the corresponding control groups.

Table 2 contains the results of the IF method applied for outlier removal.

**Table 2.** Results of IF application for the initial THz dataset.

| Group Name | Number of Outliers | Total Count of Samples | Outlier Percentage |
| --- | --- | --- | --- |
| U87-1 | 1 | 5 | 20% |
| U87-2 | 1 | 10 | 10% |
| U87-3 | 2 | 7 | 28% |

The highest percentage of outliers was observed in the THz data for the third week of the experiment. The number of outliers did not correlate with the glioma stage. After outlier removal, the data were standardized by subtracting the mean and normalizing to the standard deviation. Then, linear kernel PCA was applied. The number of principal components was the maximum available and was determined by the number of samples in each class. For example, there were seven principal components (PCs) for the U87-3 group. Hereinafter, PC1 is the first principal component, PC2 is the second, etc. The results of the outlier removal and PCA implementation are shown in Figure 8. The data for PC1 and PC4 in the third week of the experiment were the most informative. The change of point positions in the principal component subspace is due to the fact that PCA was applied to a reduced THz spectral matrix after outlier removal.

In our previous work we reported positive test results for application of SVM for THz spectral data [16], so we expected good results in this study also. The comparison of the performance of RF, XGBoost, and SVM for THz spectra analysis of glioma is a subject of further research, to be published separately. We excluded artificial neural networks (ANN) from the consideration, because it is difficult to perform important feature selection in such cases: the only working option is the Shapley values method with SHAP algorithm, but this has limitations [42]. Furthermore, for a small dataset, ANN can generate an overfitted model, being too complex with poor data generalization ability. A linear kernel SVM with default parameters was applied to data transformed into the PC feature space. Data model validation was performed by 10-fold CV with averaging and randomly balanced partitioning relative to the size of the original classes, because the number of samples in each class did not exceed 10. The quality of the predictive data model was estimated using the averaged specificity, sensitivity, accuracy (Table 3), and ROC-AUC analysis (Figure 9). Sensitivity is a measure of how well a data model describes true positive cases (i.e., tumor predicted as tumor), while specificity is a similar measure for true negative cases (i.e., non-tumor predicted as non-tumor). Accuracy describes a combination of random and systematic observational error: high accuracy means both high precision and high trueness [43]. It should be noted that, in the case of highly imbalanced class sizes, measurement of classifier performance by balanced accuracy and Matthew's correlation coefficient is recommended [44]. In our case, the size of each class was the same, so we opted to use sensitivity, specificity, accuracy, and ROC-AUC metrics. The extracted informative principal components were derived from the proximity criteria of the reference vector in the SVM to the separating hyperplane. As mentioned, after determining the informative principal components by analyzing the load matrix, informative frequencies were found.

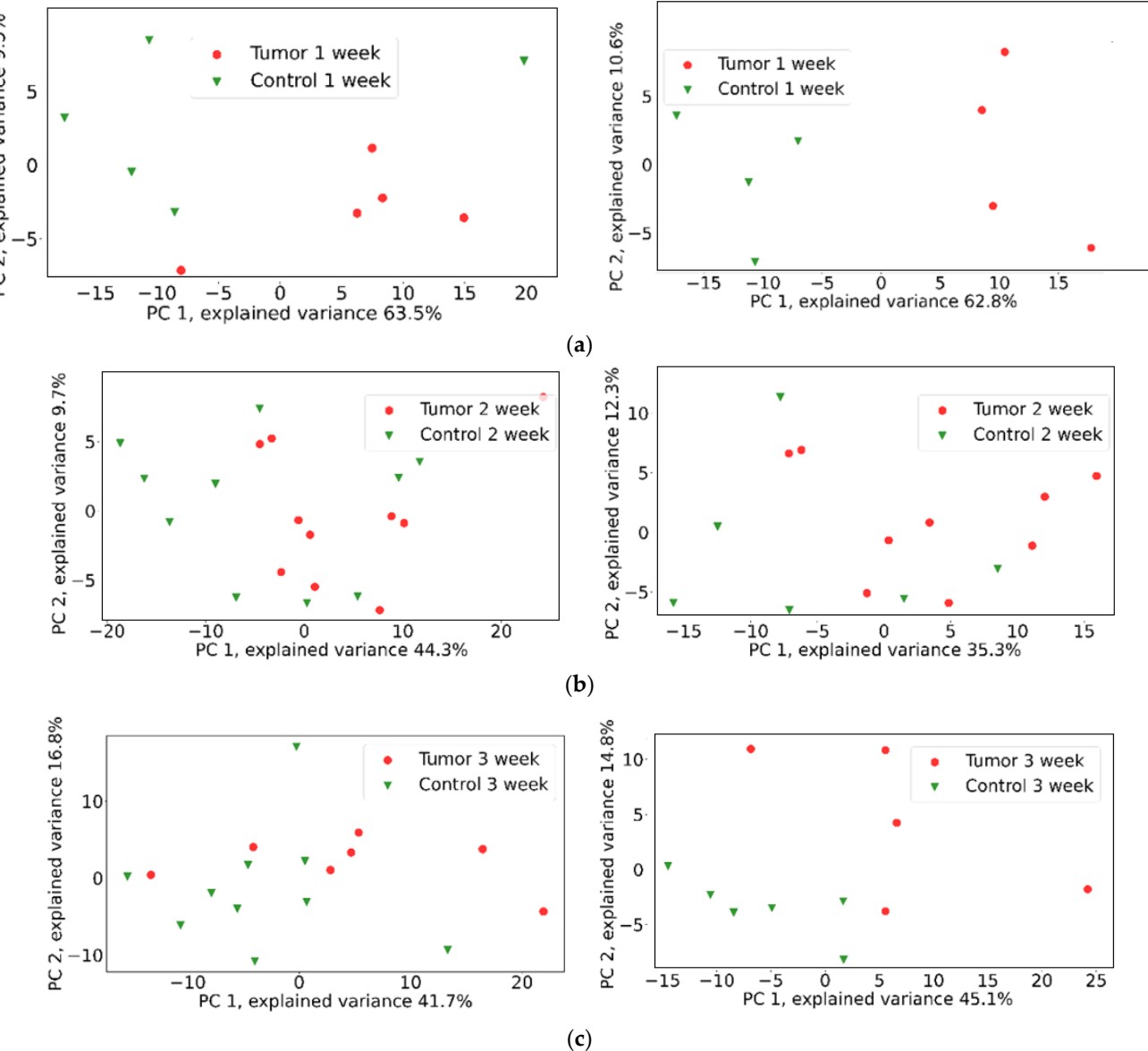

**Figure 8.** Results of PCA analysis of THz spectra for (**a**) the first, (**b**) second, and (**c**) third weeks of the experiment, (**left**) without and (**right**) with outlier removal.

**Table 3.** Average performance of the SVM classifiers of control vs tumor groups for the first, second and third weeks of the experiment, with and without outlier removal.

| Week | Outliers Removed | Sensitivity | Specificity | Accuracy |
|------|------------------|-------------|-------------|----------|
| 1 | No | 0.43 | 0.69 | 0.57 |
| 1 | Yes | 1 | 1 | 1 |
| 2 | No | 0.43 | 0.57 | 0.50 |
| 2 | Yes | 0.65 | 0.83 | 0.76 |
| 3 | No | 0.7 | 0.88 | 0.82 |
| 3 | Yes | 0.85 | 1 | 0.93 |

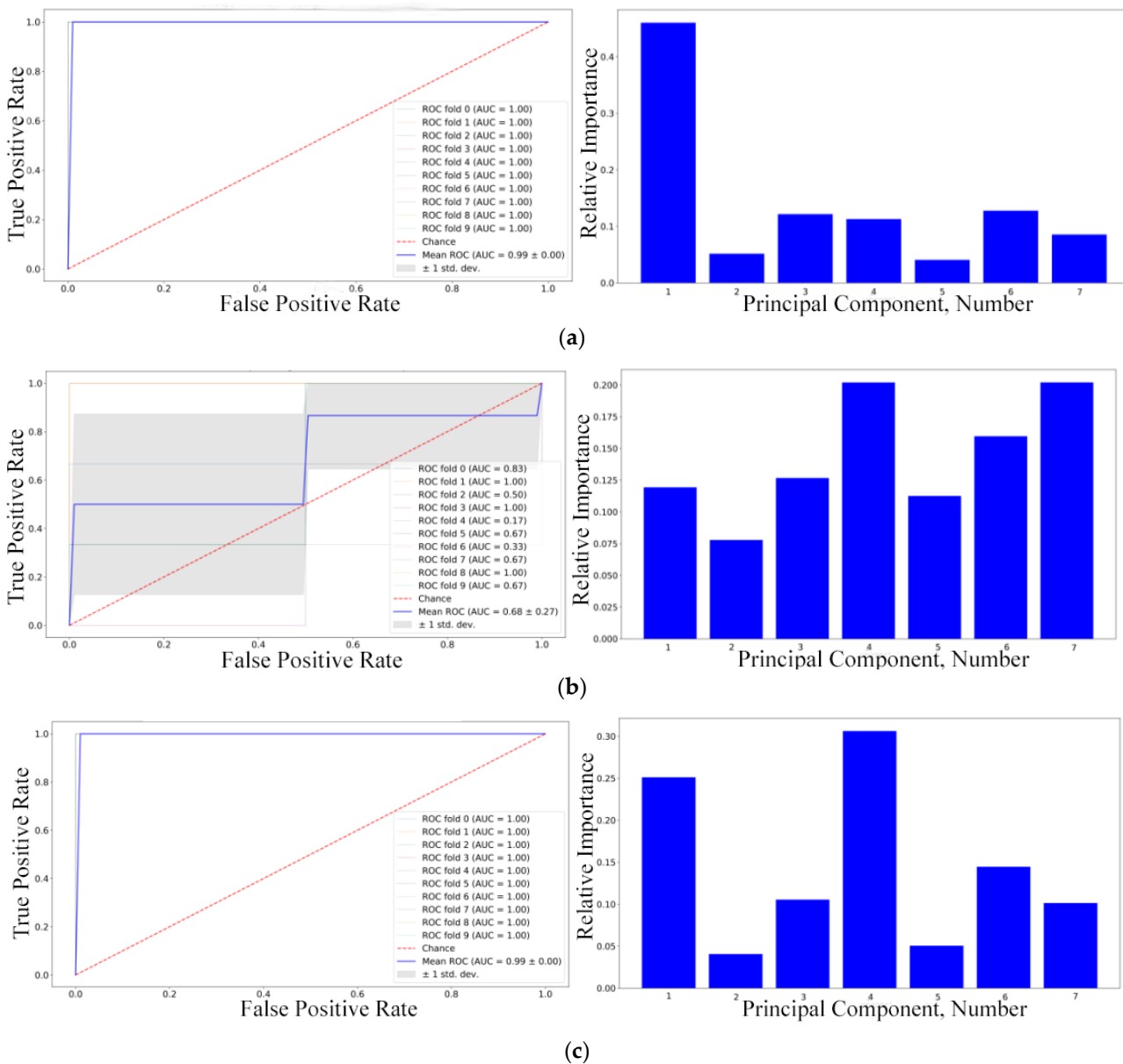

**Figure 9.** ROC-AUC analysis and informative principal components chosen by SVM for (**a**) the first, (**b**) the second, and (**c**) the third weeks of the experiment, respectively.

It can be seen from Figure 9 that, according to SVM classifiers, the most informative principal component for the first week was PC1, those for the second week were PC4 and PC7, and for the third week PC4 and PC1. It should be mentioned that the SVM accuracy and ROC-AUC values were low for the second week, so those particular informative principal components should not be trusted. With the informative principal components identified using linear SVM, the next step was to analyze the loadings matrix (see Figure 10). The discontinuous cutoff lines in Figure 10 indicate the threshold values: peaks above the positive cutoff line and below the negative cutoff line are considered informative.

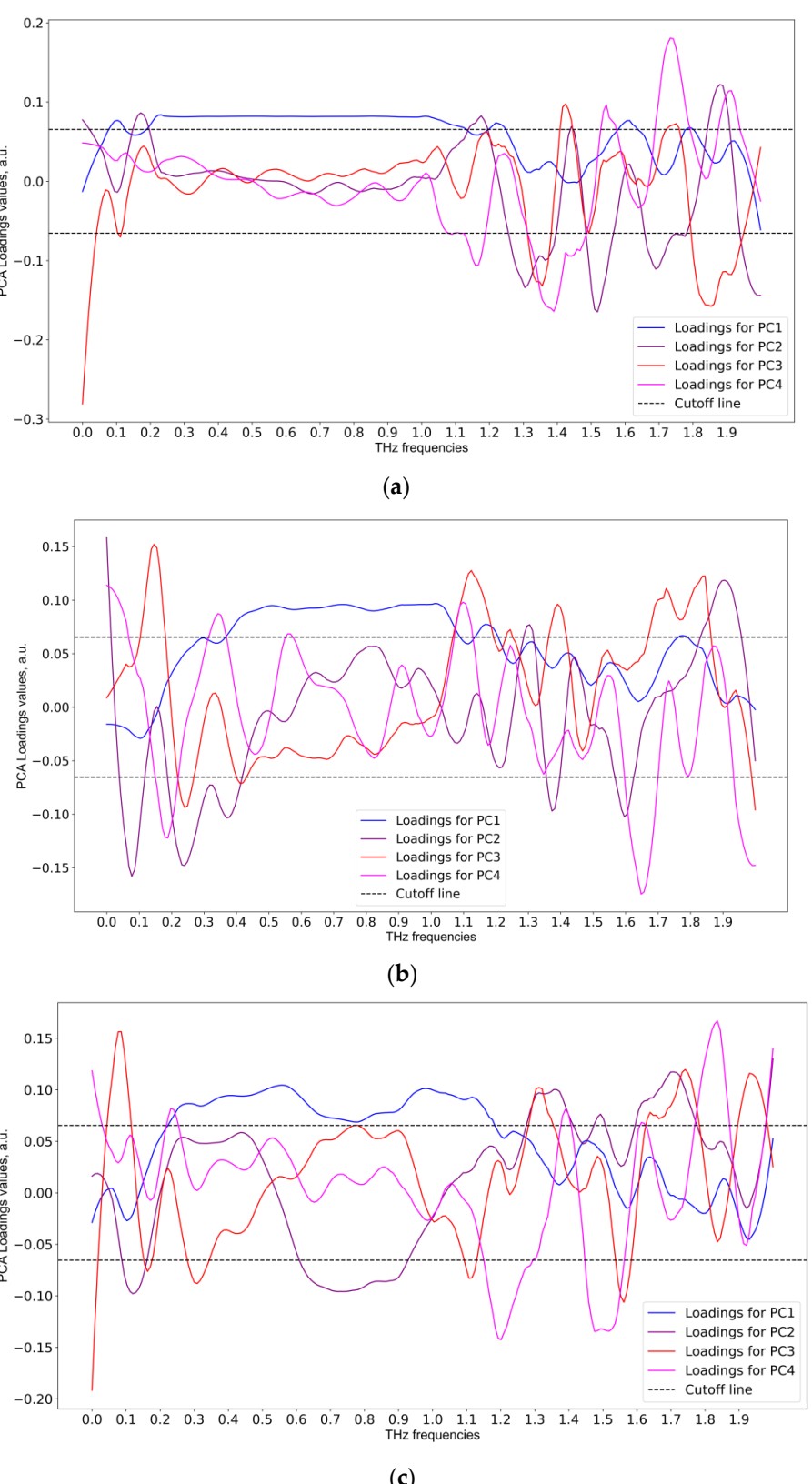

**Figure 10.** PCA loadings matrix analysis for informative feature selection, based on THz data: (**a**) first week, (**b**) second week, and (**c**) third week of the experiment.

As shown in Figure 9, the informative frequencies for the first week were 0.11, 0.22, 1.2, 1.6 THz for PC1. For the second week, the informative frequencies were 0.18, 0.35, 1.1, 1.65 THz for PC4. For the third week, the informative frequencies were 0.56, 1.0 THz for PC1, and 0.22, 1.2, 1.48, 1.52, 1.82 THz for PC4. It should be noted that the predictive data model for the second week of the experiment was of inferior quality, so the informative frequencies in that particular case cannot be considered reliable.

According to the sample size, we applied Mann–Whitney U testing (*p*-value threshold >0.5, the larger the better) for the selected informative features for the first and the third weeks of the experiment. The results reflected the known problem of using *p* value for informative feature selection in high dimensional cases [45,46]; for example, 0.08 THz was found informative, *p*-value = 0.81, but ML methods discarded it. It can be seen from the THz spectra plots that the peak at 0.08 THz had large overlapping variance and cannot be considered important. Vice versa, meanwhile, the peak at 0.56 THz had a larger difference in mean and low STD values, but a low *p*-value = 0.01. For this reason we do not apply statistics-based informative feature selection in this paper.

### 3.2. Study of Relationship between Tumor Size and THz Spectral Profile of Blood Serum by LASSO Regression Method

The parameter of regularization $\alpha$ in the LASSO method is critically important, the accuracy of the model depends on the quality of its choice. We implemented a procedure of automatic computation of optimal $\alpha$ value based on the k-fold CV with repetition (sklearn library, RepeatedKFold module with parameters n_splits (number of partitions) 5, n_repeats (number of repetitions) 5) and the LassoCV procedure from the sklearn library, with the parameters: $\alpha$ in the range 0 to 1 with step 0.01, maximum number of iterations 100,000, error during optimization 0.01.

The calculated optimal $\alpha$ value was then used to create a model, using the Lasso module of sklearn with the same error parameters and number of iterations as during optimization. The model was validated using the k-fold CV with repetition procedure described above, with the same parameters, but the sample composition differed due to applying a random number generator. The THz spectra of groups with glioma from the first to the third week were used as input data. The model quality was assessed by the determination score ($R^2$), and mean absolute error (MAE) (see Table 4). The model with $R^2$ = 0.83 was considered good.

**Table 4.** Values of MAE, variance, and optimal regularization.

| $R^2$ | MAE | $\alpha$ Value |
|:---:|:---:|:---:|
| 0.83 | 1.120 | 0.09 |

A list of informative THz frequencies was obtained (Figure 11).

As can be seen from Figure 11, the informative frequencies for the THz spectra were 0.19 and 1.9 THz, the latter making the largest contribution. These informative frequencies were similar to those obtained by the combination of PCA and SVM, especially for the third week of the experiment. Therefore, we believe that there was a relation between tumor size and THz spectra of the studied samples.

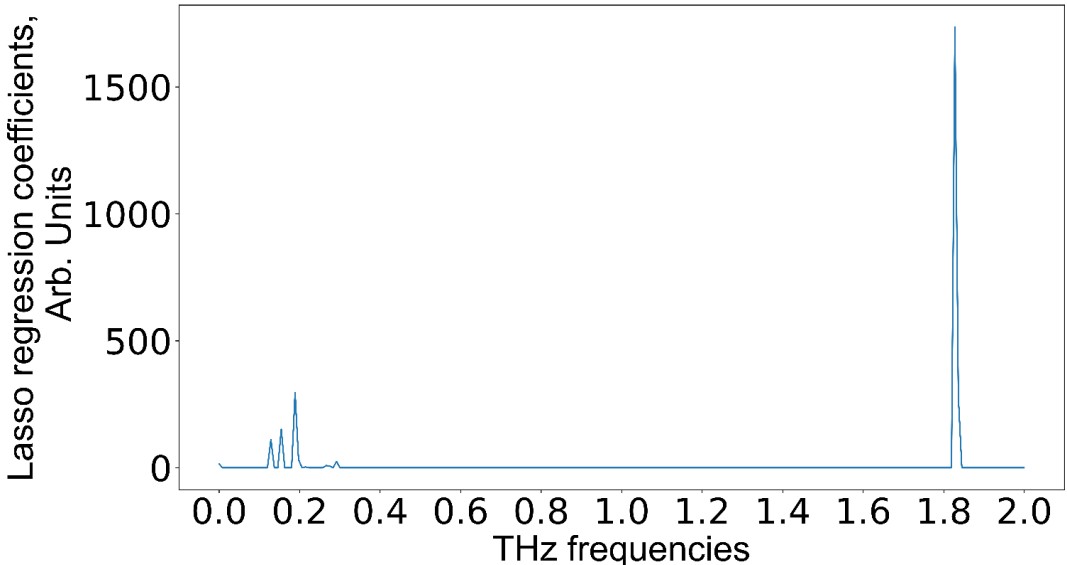

**Figure 11.** Informative frequencies for THz spectra obtained from the LASSO method.

### 4. Conclusions

We applied THz-TDS and ML methods to study the blood serum of mice in the dynamics of experimental U87 glioblastoma development. Larger tumor growth was observed from 14 to 21 days after injection of U87 cells into the subcortical brain structure.

The ML pipeline included THz spectra smoothing using the Savitsky–Golay filter, outlier removing using IF, subtraction of the mean and normalizing data to the standard deviation, informative feature extraction, and dimensionality reduction by PCA. The predictive data model was created using SVM with a linear kernel. The proposed ML pipeline provided over 90% sensitivity, specificity, and accuracy of glioblastoma detection.

The relationship between tumor size and THz spectral profile of blood serum was studied by LASSO regression. We established that serum THz absorption decreased when tumor size increased. The informative THz frequencies providing the best classification were close to 0.19 and 1.9 THz, the latter spectral region making the largest contribution. The presented results demonstrate that the ML-based blood-serum THz spectra predictive data model can allow researchers to distinguish between cancer and healthy groups, and to differentiate glioma stages. The average performance of the SVM classifiers was shown to increase after outliers were removed. The achieved efficiency of the predictive data model can be improved further by increasing the dataset volume.

**Author Contributions:** Conceptualization, O.C. and Y.K.; methodology, Y.K. and D.V.; software, D.V.; validation, D.V.; formal analysis, D.V., A.K., O.S., I.R. and E.Z.; investigation, A.K., O.S., I.R., and E.Z.; writing—original draft preparation, O.C., D.V. and A.K.; writing—review and editing, O.C., Y.K., M.K., O.S., E.Z. and A.S.; visualization, D.V. and A.K.; supervision, A.S.; project administration, O.C. and Y.K.; funding acquisition, O.C. All authors have read and agreed to the published version of the manuscript.

**Funding:** The study was funded by Russian Foundation for Basic Research and National Natural Science Foundation of China according to the research project No. 19-52-55004 using the equipment of the Center for Genetic Resources of Laboratory Animals, Institute of Cytology and Genetics, Siberian Branch of the Russian Academy of Sciences, supported by the Ministry of Science and Higher Education of the Russian Federation (unique identifier of the project: RFMEFI62119X0023). This work was supported by the Ministry of Science and Higher Education of the Russian Federation within the State assignment FSRC "Crystallography and Photonics" of the Russian Academy of Sciences, within the State assignment Institute of Laser Physics, Siberian Branch of the Russian Academy of Sciences. This work was supported by the Interdisciplinary Scientific and Educational School of Moscow University "Photonic and Quantum Technologies, Digital Medicine". The analysis

of THz data by D.V., A.K. and Y.K. was supported by the Ministry of Science and Higher Education of the Russian Federation (V.E. Zuev Institute of Atmospheric Optics of Siberian Branch of the Russian Academy of Sciences). The work of Y.K. was supported by the Tomsk State University Development Program (Priority-2030).

**Institutional Review Board Statement:** The study was conducted according to the guidelines of the Declaration of Helsinki, and approved by the Inter-Institutional Commission on Biological Ethics at the Institute of Cytology and Genetics, Siberian Branch of the Russian Academy of Sciences (Permission #78, 16 April 2021).

**Informed Consent Statement:** Not applicable.

**Data Availability Statement:** The data presented in this study are available on request from the corresponding author.

**Conflicts of Interest:** The authors declare no conflict of interest. The funders had no role in the design of the study; in the collection, analyses, or interpretation of data; in the writing of the manuscript; or in the decision to publish the results.

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
