# Peer review of "Analysis of Mouse Blood Serum in the Dynamics of U87 Glioblastoma by Terahertz Spectroscopy and Machine Learning"

_applsci, doi:10.3390/app122010533_

Round 1

Reviewer 1 Report

In this manuscript, the authors propose a machine learning process to evaluate the blood serum of mice with different growths of glioblastomas. Authors state that it is possible to distinguish between cancer from healthy groups and diagnose different glioma stages using their procedure.  

I would like to make the following inquiries and comments to the authors before the manuscript could be considered for publication:

1.     As the journal is multidisciplinary, I think readers would benefit if more detail is included in the techniques and procedures described in the manuscript.  In particular, more information should be included in the following subjects:

a.     Nature of glioblastomas (what are they?)

b.     What is metabolomics (as not all readers may have a medical or biological background)?

c.     How is the THz data obtained from the time domain? 

d.     Why should ML “should” be applied to THz spectra with no sharp peaks, as the authors state in line 55 of the Introduction?

e.     How do the authors determine the handling of blood serum samples?  At what temperature are the samples when measured with the THz spectrometer?  The temperature is constant during all measurements?  Can this be a factor that changes the results?

f.      Why is THz radiation collimated and not focused in the sample?

g.     What are the definitions of sensitivity, specificity, and accuracy presented in table 3?

2.     There is not much information about the kind of material used for printing the cuvettes.  The references are not directly related to the THz characteristics of Watson material.  I would suggest the authors to include what are the material's characteristics and the printing process.  There is some absorption of the cuvette, is this absorption the same for all printed cuvettes? How variable is that parameter?  Would the variability of this change results? How controlled are the thickness and roughness of the cuvettes?

3.     Another important concern is that there is not much description of how and why the tumor growth causes changes in the blood composition and how these changes are related to the changes THz signal.  What is physically, optically, and biologically happening with the transmission of the THz signal?  

4.     In order to be able to conclude that the method is really a candidate for diagnosis, the results should also be compared with a gold standard.

There is many fundamental explanations that should be included in the manuscript before it can be considered for publication, especially with the important conclusions that are presented.

Author Response

Dear Reviewer,

We would like to thank you for your work with our submission and for your thoughtful and relevant remarks. We have revised our manuscript according to your comments. Please, see the details of this revision below. For your convenience, the main changes are highlighted with the green-colored text in the revised manuscript.

Our replies are in the attached file.

Reviewer 2 Report

The manuscript entitled: “Analysis of mouse blood serum in the dynamics of U87 glioblastoma by Terahertz spectroscopy and Machine learning” by Denis Vrazhnov, et al, created a prediction data model based on terahertz time-domain spectroscopy and machine learning which allows to differentiate cancer and healthy cells. The work is sound and the results are remarkable. However, the abstract should be checked thoroughly. Overall, the work is certainly publishable and contributes to research area of glioblastoma detection. It could be published in the special Issue "Applications of Millimeter-Wave and Terahertz Technologies” of Applied Sciences after minor revisions as follow.

1)    Abstract should be rewritten. The information given in the abstract is correct and precise, but there is no connection between the sentences. This lack of connection provokes the loss of the reader throghout the text.

2)   Reference number 15 is not mentioned throghout the text and in line 46 reference number 14 appears twice one after the other. Please, check it. I think that the second reference of number 14 in line 46 is actually the reference number 15. Review also line 216.

3) The material of the cuvette used for the experiments should be mentioned in section 2.2.

4)   The first sentence of the Section 2.3 (lines 147 and 148) does not make sense. It should be rewritten.

5)   In the explanation of Figure 4 “orange dot is a rare data, but still not an anomaly” is written, but there is not any orange dot in the figure. Which is the meaning of blue dots?

6)  The quality of the images from Figure 6 is quite poor (specailly in the axis). Consequently, the monitoring of the experiment is hard for the reader. Exporting the graphics in “.tiff” mode could be a better option. Same explanation for the images from Figure 7.

7)    The quality of the images form Figure 8 is correct, but I should increase the size of the letters from the axis for a better understanding.

Author Response

Dear Reviewer,

We would like to thank you for your work with our submission and for your thoughtful and relevant remarks. We have revised our manuscript according to your comments:

1) Abstract has been rewritten.

2) Reference numbers have been corrected.

3) The material of the cuvette used for the experiments has been be mentioned in section 2.2.

4) The first sentence of the Section 2.3 has been rewritten.

5) The caption for Fig. 4 (now Fig. 5) has been corrected.

6) We have made the necessary corrections in Fig. 6 and 8 (now Fig. 7 and 9).

For your convenience, the main changes are highlighted with the green-colored text in the revised manuscript.

Reviewer 3 Report

This paper employs Terahertz spectroscopy and  machine learning to discriminate between control and tumor groups pertaining to glioblastoma of different time frames. Experimental results show the good results of SVM with linear kernel in tackling this prediction task. The paper is well-described, promoting a good application to SVM. However, the following comments need to be addressed:

-Reporting results using other machine learning algorithms such as neural networks, XGBOOST, and Random Forests
-in Table 3, reporting results using additional performance measures such as balanced accuracy, and Matthews correlation coefficient
-Employing a statistical test to report the learning method generating significant results.

Author Response

Dear Reviewer,

We would like to thank you for your remarks and comments. We have revised our manuscript according to your comments. Please, see the details of this revision below. For your convenience, the main changes are highlighted with the green-colored text in the revised manuscript.

Comment #1: Reporting results using other machine learning algorithms such as neural networks, XGBOOST, and Random Forests.

Authors' Response:  In our previous work we already reported the good results of application of SVM for THz-TDS [Konnikova, Maria R., et al. "Malignant and benign thyroid nodule differentiation through the analysis of blood plasma with terahertz spectroscopy." Biomedical optics express 12.2 (2021): 1020-1035]. We also preparing another paper, where we compare XGBoost, Random Forest and SVM methods applied to THz spectroscopy along with the informative feature selection. It should be mentioned, that it is difficult to perform important feature selection for neural networks, only working option is Shapley values, but it requires a large number of computations. Also, Randopm forests, XGBoost and neural networks creates complex models and tend to overfitting on a small dataset like ours, which will be also presented in the next paper.

Comment #2: in Table 3, reporting results using additional performance measures such as balanced accuracy, and Matthews correlation coefficient.

Authors' Response:  The use of balanced accuracy and Matthews correlation coefficient is justified, when sample size is very different [Zhu, Qiuming. "On the performance of Matthews correlation coefficient (MCC) for imbalanced dataset." Pattern Recognition Letters 136 (2020): 71-80.], which is not in our case. Besides, we use balanced training option, when classes have significant difference in size. Our results can be also verified by presented ROC-AUC graphics with K-fold splitting.

 Comment #3: Employing a statistical test to report the learning method generating significant results.

Authors' Response:  According to the sample size we applied Mann-Whitney U test (p-value threshold is >0.5, the bigger the better) for the selected informative features for the first and the third week of the experiment. Second week was not tested, because we failed to select reliable IF from the data. The results correlate with known problem of using P-value for informative features selection in high dimensional case [1, 2]: for example, 0.08 THz was found informative, p-value = 0.81, but machine learning methods discards it. It can be seen from the THz spectra plots, that peak at 0.08 THz has large overlapping variance and cannot be considered as important. And vice-a-versa, peak at 0.56 THz has larger difference in mean and low in STD values, but has low p-value=0.01). That is why we do not the statistics based informative features selection in this paper.

  1. Goodman, S. N. Toward evidence-based medical statistics. 1: The P value fallacy. Annals of internal medicine 1999, 130 (12), 995-1004. doi: 10.7326/0003-4819-130-12-199906150-00008.
  2. Halsey, L.G.; Curran-Everett, D.; Vowler, S.L.; Drummond, G.B. The fickle P value generates irreproducible results. Nature methods 2015, 12 (3), 179-185. doi: 10.1038/nmeth.3288.

An appropriate correction has been made in the text (see Line 316-326).

Round 2

Reviewer 1 Report

I thank the authors for taking into account previous observations.  I believe most of the concerns have been addressed.  

Author Response

Dear Reviewer,

We would like to thank you again for your work with our submission and for your thoughtful and relevant remarks. We have revised our manuscript according to your comments. We checked English language and style and made the appropriate corrections. Please, see the details of this revision below. For your convenience, the main changes are highlighted with the yellow-colored text in the revised manuscript.

Reviewer 3 Report

Authors have improved the manuscript. However, two comments, "Reporting results using other machine learning algorithms such as neural networks, XGBOOST, and Random Forests." and "in Table 3, reporting results using additional performance measures such as balanced accuracy, and Matthews correlation coefficient." have not been addressed properly.

Author Response

Dear Reviewer,

Thank you very much for reviewing our paper and very useful suggestions. We have revised our manuscript according to your comments. Please, see the details of this revision below. For your convenience, the main changes are highlighted with the yellow-colored text in the revised manuscript.

 Comment #1: Reporting results using other machine learning algorithms such as neural networks, XGBOOST, and Random Forests.

Authors' Response

In our previous work we already reported the positive tests of application of SVM for THz spectral data [16], so we expected good results it this study too. The comparison of the performance of RF, XGBoost, and SVM for glioma THz spectra analysis is a subject of further research and a separate publication. We are now preparing another paper where we will demonstrate comparative analysis of XGBoost, Random forest and SVM for the informative feature selection in THz spectral data.

We exclude artificial neural networks (ANN) from the consideration, because it is difficult to perform important feature selection for them: the only working option is Shapley values method and SHAP algorithm, but it has limitations [42]. Also on a small dataset, ANN can generate an overfitted model (too complex with poor data generalization ability).

An appropriate correction has been made in the text (see Line 265-273).

We have added additional reference

  1. Kumar, I. E., et al. Problems with Shapley-value-based explanations as feature importance measures. International Conference on Machine Learning PMLR 2020, 5491-5500.

Comment #2: in Table 3, reporting results using additional performance measures such as balanced accuracy, and Matthews correlation coefficient.

Authors' Response:  The quality of the prediction data model was estimated, using the averaged specificity, sensitivity, accuracy (Table 3) and ROC-AUC analysis (Figure 9). It should be pointed out that, in the case of highly imbalanced size of the classes, it is recommended to measure classifier performance by balanced accuracy and Matthew’s correlation coefficient [44]. In our case, the size of the classes is practically the same, so we decided to use sensitivity, specificity, accuracy and ROC-AUC metrics.

An appropriate correction has been made in the text (see Line 283-286).

We have also added additional reference:

  1. Zhu, Q. On the performance of Matthews correlation coefficient (MCC) for imbalanced dataset. Pattern Recognition Letters 2020, 136, 71-80. doi: 10.1016/j.patrec.2020.03.030.
